# Validity of Three Survey Questions for Self-Assessed Sedentary Time

**DOI:** 10.3390/ijerph19074269

**Published:** 2022-04-02

**Authors:** Viktoria Wahlström, Mikael Nygren, David Olsson, Frida Bergman, Charlotte Lewis

**Affiliations:** Department of Public Health and Clinical Medicine, Umeå University, 901 87 Umeå, Sweden; mickenygren91@gmail.com (M.N.); david.olsson@umu.se (D.O.); frida.bergman@umu.se (F.B.); charlotte.lewis@regionvasterbotten.se (C.L.)

**Keywords:** sensor-based measurements, self-reported, sedentary behavior, sitting, questionnaire

## Abstract

Time spent in sedentary behavior (SB) has increased during the last decades. Accurate assessments are of importance when studying health consequences of SB. This study aimed to assess concurrent validity between three different questions for self-reported sitting and thigh worn accelerometer data. In total, 86 participants wore the ActivPAL accelerometer during three separate weeks, assessing sitting time with different questions each week. The questions used were Katzmarzyk, GIH stationary single-item question (SED-GIH), and a modified version of the single-item from IPAQ short form. In total 64, 57, and 55 participants provided valid accelerometer and questionnaire data at each time-point, respectively, and were included for analysis. Spearman and Pearson correlation was used to assess the validity. The three questions, Katzmarzyk, SED-GIH, and a modified question from IPAQ all showed a weak non-significant correlation to ActivPAL with r-values of 0.26, 0.25, and 0.19 respectively. For Katzmarzyk and SED-GIH, 50% and 37% reported correctly, respectively. For the modified IPAQ, 53% over-reported and 47% under-reported their sitting time. In line with previous research, our study shows poor validity for self-reported sitting-time. For future research, the use of sensor-based data on SB are of high importance.

## 1. Introduction

It is well known that physical activity (PA) has positive health benefits. Studies have shown that high levels of physical activity, regardless of intensity, combined with less time spent sedentary reduce the risk of premature death [1]. Sedentary behavior (SB) is commonly defined as any waking behavior in a sitting, reclining, and lying posture with an energy expenditure of ≤1.5 metabolic equivalents [2]. In the modern western society, time spent in SB activities, such as watching television, sitting at computers, and passive commuting to work, have increased during the last decades [3].

In order to measure the amount of SB and PA, either subjective measurements (questionnaires) or sensor-based measurements with wearable devices, such as accelerometers and heart-rate monitors, are used [4]. Subjective measurements are a simple and cost-effective way to gather data from larger populations. Sensor-based measurement are more precise [5]; however, the resources for collecting and processing the sensor-based data are more expensive and burdensome [6]. 

SB can be assessed using hip- or thigh worn accelerometers. Since a hip-worn accelerometer does not provide the opportunity to differentiate between standing and sitting time, there is a risk of misclassification of time spent sitting and standing. A thigh-worn inclinometer on the other hand has shown to differentiate between different postures with higher accuracy, and is, therefore, considered the golden standard for measuring sedentary time [5,7,8,9,10].

Several studies have compared self-assessed questionnaire data with sensor-based data from hip-worn accelerometers or thigh-worn inclinometers [11,12,13,14,15]. Studies have shown that people often estimate their time spent sedentary incorrectly compared to sensor-based measurement. This can be a result of misunderstanding the questions as well as recall bias influenced by social and cultural norms [3]. It has been suggested that future studies should compare questions to sensor-based measurements to further improve the development of subjective measurements [16]. It can, however, be beneficial to compare different questionnaires in the same population, in order to investigate if there are methods that have a higher precision than others, and if the patterns of possible misclassifications differ between questionnaires. As subjective measurement methods have many advantages, it is of great interest to find questions with high validity and reliability. The validity of questionnaires can be determined by comparing questionnaire assessments to sensor-based measurements. The aim of this study was to evaluate the concurrent validity of three different questions used to assess sedentary time in office workers using thigh worn accelerometer as reference. 

## 2. Materials and Methods

### 2.1. Setting and Recruitment

The project is based on data collected within the Active Office Design (AOD) study, a longitudinal quasi-experimental study among office workers in a Swedish municipality. The overarching aim of the AOD study was to evaluate the effects of different office types on work environment, productivity, health, SB, and PA [17].

In total, the AOD study involved 371 employees, of whom, 59% relocated from a cell office to a flex office, and 41% relocated from a cell office to another traditional cell office. Among these, 86 participants (43 from each office type) were recruited for repeated sensor-based measurements of SB, PA, and body measures. The studied organization provided the researchers with lists of employees involved in the relocation process. Within these lists, a computer-generated list of random numbers within each office type was prepared by a researcher not otherwise involved in the study. Following this list, selected employees were sent an e-mail invitation of the study. The e-mail was followed by a phone interview. To be included in the study, participants should be (1) 18–63 years of age, (2) working 32 h a week or more, (3) spending more than 60% of work hours inside the office, and (4) not planning to relocate to another worksite during the study period. Recruitment was performed between September and December 2014, and all participants signed an informed consent. In the original study, the sensor-based measurement of SB and PA were performed twice before relocation (6 and 12 months), and at three timepoints after relocation (6, 11, and 18 months). During the ongoing measurements, we also collected data on self-rated sitting time at the three timepoints after relocation, hereinafter referred to as measurement 1, 2, and 3. Data for the current study was collected between November 2015 and March 2017. The flex office group relocated 6 months before the cell office-group, which means that the measurements were carried out with a seasonal difference between the groups.

Parallel with the office relocation, a multicomponent intervention was implemented in the organization. The intervention aimed to decrease SB and increase PA among the employees, both during work hours and leisure time. In short, the intervention program included components targeting both organizational, environmental, and individual levels. Intervention activities were (1) lectures aiming to increase awareness of the relationship between SB, PA, and health; (2) workshops for managers; and (3) communication campaigns encouraging employees to break up prolonged sitting and to vary between sitting av standing. The campaigns also highlighted the importance of everyday PA, like active commuting, taking the stairs, and/or using treadmill workstations available at the workplace. The intervention program is described in more detail elsewhere [18,19].

### 2.2. Background Characteristics

Background characteristics for age, general health, managerial position, and exercise habits was collected via questionnaires distributed to all employees at timepoint 1 and 3. Short Form 36 (SF-36) is a questionnaire constructed to survey health status in medical studies [20,21]. For assessment of self-rated health, we used one question from SF-36, where participants estimated their health on a five-graded scale from “bad to excellent” [21]. Self-reported exercise was assessed using the question “how many days during the past three months have you exercised in workout clothes, with the purpose of improving your fitness and/or to feel good” on a five-graded scale ranging from “never to >3 times per week” [22]. Body measurements were performed at the workplace at measurement 1 and 3. The participants wore underwear during the measurements. Body height was measured to the nearest 0.1 cm with a wall-mounted stadiometer (Hyssna 4146, Measuring Equipment AB, Hyssna, Sweden), and body weight to the nearest 0.1 kg using a calibrated electronic digital scale (Tanita BWB-800 MA; Umedico AB, Rosersberg, Sweden). BMI was calculated as weight (kg) divided by height (m) squared. 

### 2.3. Sensor-Based Measurements of Sedentary Behavior

Sitting time was measured using ActivPAL, with the participants wearing the sensor on their right thigh for 24 h per day for a week (PAL Technologies Limited, Glasgow, UK; default settings). During the weeks of measurements, participants noted in a logbook what time they got up and went to bed, whether it was a work or non-workday, and periods of non-wear time. The logbook was used to distinguish total time for workdays and non-workdays. ActivPAL has shown high validity in terms of distinguishing sitting/lying from standing and stepping, as well as transitions between postures [9,23]. Data for ActivPAL were processed using a custom-made excel macro (HSC PAL analysis software v2.19s). For a measurement period to be eligible for analysis, it had to include at least three workdays and at least one non-workday. To be included, a measurement day needed to include >10 h of data [9]. If there were more than seven days of eligible data for a measurement period, the first five valid workdays and first two valid non-workdays were used in the analysis. For ActivPAL-data, time in SB was calculated as the sum of SB on all valid days divided by the number of valid days. 

### 2.4. Self-Reported Sedentary Time

At measurements 1, 2, and 3, participants were asked to assess their total sitting time for the week of measurement using different questions at each measurement (Figure 1). At measurement 1, a question developed by Katzmarzyk was used [24]. The question was formulated “How much of your waking time do you spend sitting” and were assessed on a five-graded scale from “sitting almost all of the time” to “almost none of the time”. At measurement 2, the GIH stationary single-item question (SED-GIH) was used with the question “How much do you sit during a normal day excluding sleep”, where sedentary time was assessed on a seven-graded scale from “almost all the time” to “never”. At measurement 3, a modified version of the single-item question from the short form of the International Physical Activity Questionnaire (mIPAQ) was used. In this question participants were asked to report the number of hours and minutes spent sitting during weekdays and weekend days separately for the last 7 days. 

### 2.5. Data Treatment and Statistical Analysis

To be included in the analysis at each timepoint, participants had to have valid data for both ActivPAL and self-reported sitting time. For measurement 1, sitting time measured by ActivPAL was converted to percent of the total waking time per day. The percent was then categorized with intervals that represented the different answer categories for the Katzmarzyk question (<21%, 21–40%, 41–60%, 61–80%, 81–100%). The percent intervals were then compared with the answer-categories. For measurement 2, sitting time in minutes measured by ActivPAL was categorized in seven different intervals to match the time expressed in hours in the scale (<30 min, 30–219 min, 220–389 min, 390–569 min, 570–749 min, 750–929 min, and >929 min). Each of these intervals represented an answer-category on the SED-GIH scale. These intervals were then compared to the answer-categories for SED-GIH. For measurement 3, the mIPAQ question, which was answered in hours and minutes per week, was converted into minutes. By using the formula, (weekday sitting time × 5 + weekend sitting time × 2)/7 we calculated a mean value for self-reported sitting per day in minutes. This value was compared to the sensor-based measurement in minutes for that time-point.

For statistical analysis we used SPSS software v. 27 (IBM Corp, Armonk, NY, USA). The percentage of agreement between the sensor based and subjective variables was reported. We used Spearman’s rank correlation, r_s_, to investigate the correlation between self-reported sitting from the Katzmarzyk and SED-GIH and sensor-based measures of sitting time. To assess the correlation between the mIPAQ and the sensor-based measures Pearson correlation, r_p_, was used.

## 3. Results

Of the initial 86 eligible participants for the study, 64, 57, and 55 provided valid data at the three time-points, respectively, and were included in the analysis (Table 1). The reasons for dropout from the study were voluntary quitting (*n* = 10), parental leave (*n* = 6), sick leave (*n* = 7), or not providing sufficient data from the sensor or the questionnaires (*n* = 8). Participants were between 28 and 64 years old. The mean age, BMI, and percentage of participants sex is presented in Table 1. For the total sample, participants had, on average, 6.9 valid days of data per week and the mean value of valid hours per day was 15 h and 45 min. More detailed information is shown in Table 1. 

There was a weak relationship (r = 0.19–0.26) between sitting time compared to ActivPAL, for all three subjective questions. It was somewhat more common to overreport sitting time (18–29%) than it was to underreport (14–26%) (Table 2).

For Katzmarzyk, the category with most overreporting, with twelve participants, was the category “3/4 of the time” and the category with most underreporting was the category “1/2 of the time”, with seven participants (Figure 2). For the SED-GIH question, the answer category with most overreporting was the category “10–12” hours and the category with most underreporting was the category “4–6 h” with seven and ten participants, respectively. No participants used the answer categories “1–3 h” and “never” (Figure 3). For measurement 3, where mIPAQ was used, 53% over-reported and 47% under-reported their sitting time. The over-reporters assessed their sitting time on average 114 min (MD 82, SD 97) higher than the sensor-based measurement, and the under-reporters assessed their sitting time 125 min (MD 120, SD 92) lower than the sensor-based measurement.

## 4. Discussion

The aim of this study was to evaluate the concurrent validity of three different questions for self-reported sitting time using sensor-based data from ActivPAL. Overall, we found that the self-reported data had poor accuracy to determine time spent in SB, and there was a weak, non-significant correlation between each of the three different questions and the sensor-based data. Katzmarzyk had the highest amount of correct reporting with 50%, meaning that half of the studied population were not able to correctly classify their time spent sitting, and indicating that the patterns of possible misclassifications differ between questionnaires.

Our results are in line with a study by Chastin et al. [25], which concludes that using self-reported data for SB will always lead to some sort of misclassification and will not provide the same accuracy and precision as sensor-based data. 

Previous studies have shown that participants tend to under-report their sitting time [26,27]. The tendency to under-report sitting time is believed to be caused by recall bias, social acceptance, and a desire to be seen as active [28]. In contrast to other studies, our study showed that slightly more participants over-reported than under-reported their sitting time. This may be due to the PA promoting intervention that took place in the workplace during the measurement period, which may have led to participant being more aware and reflective over their sitting time and might have made the participants less prone to underestimate their sitting. In a study by Dollman et al. [29] comparing sitting time between Australian farmers and office-workers, the authors argued that since desk-based occupations are relatively highly regimented with regular breaks for coffee and lunch, office-workers may be better equipped to recall their workplace sitting as they can more easily draw awareness upon their “typical” workday. 

For all the questions, there were slightly more over-reporters than under-reporters. When it comes to the mIPAQ, a study by Chastin et al. [30] showed contrasting results where more people under-reported their sitting time than over-reporting. This may be due to the differences in background characteristics between our studies. However, Clark et al. [12] showed no difference in terms of characteristics (gender, education, and BMI) by those who under-reported their sitting time compared to those who over-reported their sitting time. Most of the previous validation studies on SB have not reported the physical activity levels of the study populations. To our knowledge, only one study considered whether participants activity level might influence the tendency to over- or under-report sitting time [15]. In a stratified analysis, Kallings et al. [15] found that persons performing more moderate to vigorous physical activity had higher probability for correct reporting of their stationary time, and they were more prone to over-report their stationary time compared to those being less physically active. Since our study population was a physically active group that averaged about 10,000 steps per day, and reported to exercise regularly, this might have influenced the tendencies to over-report sitting time. Since the activity level of the participants may impact the self-reported data, it might be beneficial for future validity studies to report the physical activity level of the study population.

For the two questions with categorical answers, Katzmarzyk and SED-GIH, few of the participants classified their sitting time in the highest or lowest categories. For Katzmarzyk, there were only four participants that self-reported in the category “almost all the time” and only one in the category “almost no time” (Figure 1). Further, the sensor-based data from that week showed that none of the participants were objectively classified in any of the highest or lowest two categories. That means, that according to ActivPAL, all of the participants were objectively classified in the three middle categories. For SED-GIH, which is a seven-graded scale, none of the participants classified their sitting time in any of the two lower categories “never” and “1–3 h”. The sensor-based data also showed that none of the participants was classified in any of those answer-categories, nor was any of the participants classified in the category “almost all day” (Figure 3). This leads to the discussion of whether the outermost categories should be combined into the same category. In a previous validation study by Kallings et al. [15], the answer categories in the SED-GIH question were reclassified into five categories during the analysis, since there were very few participants who reported sitting ”virtually all day” and “never”. The authors argue that the use of the outermost categories could be of value for participants when answering the question, as the verbal anchors “virtually all day” and “never” could make it easier for participants to relate compared to “<1 h” and “>15 h”. However, the categories could then be merged when analyzing the data. 

Our study contributes to the understanding of the difficulties with the development of questionnaires that possess a high validity to measure SB when comparing to sensor-based data. A weakness in this study is that a relatively small and homogenous group of physically active office workers constituted the study population. Age has previously shown to introduce bias in self-reported SB [15,31], but, due to the small sample size in our study, no age-related analysis was conducted. The generalizability of our results on a broader population may therefore be limited. The repeated measurements of SB in the study provided a novel possibility to collect parallel data on self-reported SB in the studied group, which is a strength. On the other hand, the ongoing intervention at the workplace might also have affected the awareness and, thereby, the self-reported SB. A strength with our study was the high quality of the sensor-based data, with a high number of valid days and hours per day.

## 5. Conclusions

In line with previous validation studies, we found that the concurrent validity was low when comparing self-reported SB to ActivPAL-based assessments for the questions by Katzmarzyk, the SED-GIH, and mIPAQ. When assessing SB on individual basis in intervention studies, sensor-based measurements should predominantly be used. Further studies including different groups of participants and activity levels should be conducted. 

## Figures and Tables

**Figure 1 ijerph-19-04269-f001:**
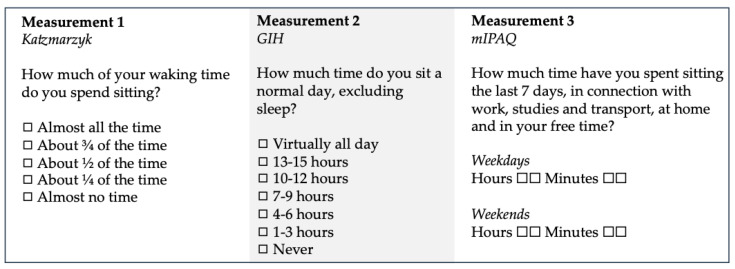
The three different questions for self-report of sedentary behavior at the three different measurements.

**Figure 2 ijerph-19-04269-f002:**
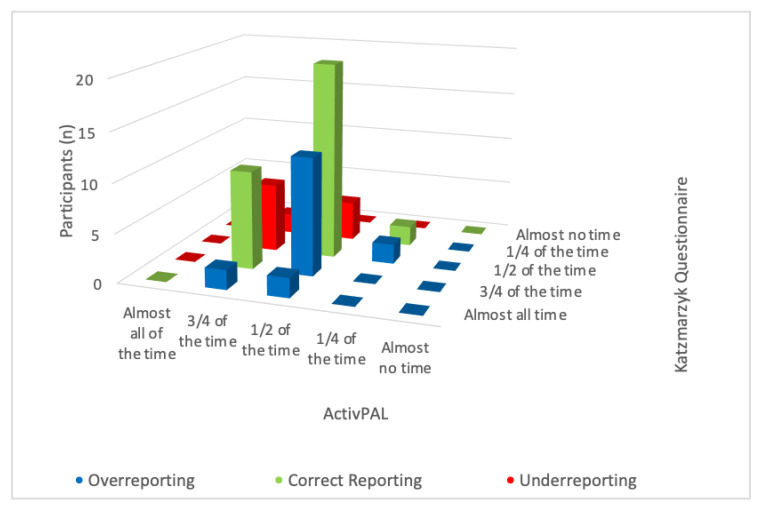
Frequency of over-reporting, correct reporting, and under-reporting in the different response categories for Katzmarzyk compared with ActivPAL.

**Figure 3 ijerph-19-04269-f003:**
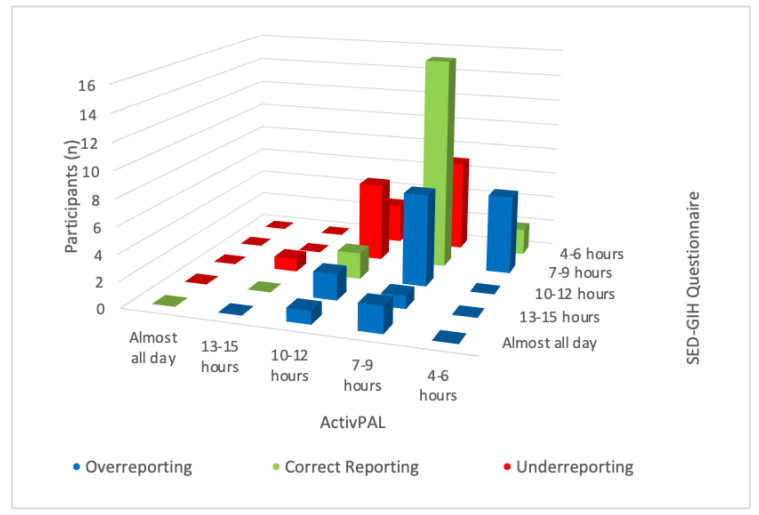
Frequency of over-reporting, correct reporting, and under-reporting in the different response categories for SED-GIH compared with ActivPAL.

**Table 1 ijerph-19-04269-t001:** Background characteristics for the three different measurement periods.

Descriptive characteristics	Measurement 1	Measurement 2	Measurement 3
Participants, *n*	64	57	55
Age, mean (SD)	50.1 (9.9)	50.4 (9.5)	50.8 (9.6)
Women, %	82.8	84.2	83.6
BMI, mean (SD)	26.3 (4.1)	26.4 (3.9)	26 (3.8)
Managers, %	15.6	15.8	16.4
Self-reported health, %			
Very good and excellent	65.6		66.1
Fairly good and bad	32.8		33.9
Self-reported exercise %			
Never	9.4		10.7
Occasionally	20.3		19.6
Once a week	12.5		14.3
2–3 times/week	37.5		39.3
>3 times/week	18.8		16.1
Measurements of SB and PA			
ActivPAL, mean (SD)			
Total wear time (number of days)	6.9 (0.04)	7.0 (0.4)	6.9 (0.4)
Wear time per day, h and min	15.39 (33)	15.81 (40)	15.49 (40)
Sedentary time, min per day	519 (96)	511 (105)	526 (93)
Steps per day	9764 (2824)	10,361 (2565)	9738 (2597)

**Table 2 ijerph-19-04269-t002:** Frequency of over-reporting (self-report > sensor-based), correct reporting (self-report = sensor-based), and under-reporting (self-report < sensor-based) for the three different questions compared to ActivPAL.

	Over- Reportingn (%)	Correct Reporting *n* (%)	Under- Reporting *n* (%)	Spearman’s Rank Correlation r_s_	*p*-Value	Pearson Correlation r_p_	*p*-Value
**Katzmarzyk**	18 (28)	32 (50)	14 (22)				
**GIH**	19 (33)	21 (37)	17 (30)	0.26	0.04		
**mIPAQ**	29 (53)		26 (47)	0.25	0.06	0.19	0.16

The associations were interpreted as, none (r <0.1), weak (r = 0.10–0.29), modest (r = 0.30–0.49), and strong (r ≥ 0.5).

## Data Availability

The full data are not publicly available due to ethical/privacy reasons. On request to the corresponding author, anonymous data are available for scientific purposes.

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
