# Peer review of "Validity of Three Survey Questions for Self-Assessed Sedentary Time"

_ijerph, 2022, doi:10.3390/ijerph19074269_

Round 1
Reviewer 1 Report
The study is sound in its method and the implications are sound and valid yet what is the novel about the study is unclear.
Author Response
Thank you for taking your time to review our paper. We believe that the novelty with our paper is the investigation of different questionnaires in the same office population, to investigate if there are methods that have a higher precision than others. We have tried to highlight this in the discussion, Page 7, Line 217.
Reviewer 2 Report
The paper focuses on the analysis of three surveys for the determination of sedentary behaviors, through a comparison with data obtained from a thigh accelerometer.
The authors concluded that self-assessment by questionnaires presents a great overestimation or underestimation of sedentary behaviors, and the results are concordant with the current literature.
The paper is clear, however, methods should be implemented.
1) Authors should specify why they selected workers employed in the relocation process.
2) In my opinion the age of subjects is very heterogeneous (18-63 years), the population and data analysis should be conducted for age categories (18-35 years; 35-50 years; 50-63 years). The presented data set does not consent to appreciate differences linked to age.
3) what means "working 75% or more"? The total of daily working hours should be specified.
4) What means SF-36? Please clarify.
The final population of the study is quite limited, as indicated by the authors in the discussion. This represents the major limitations of the study together with the heterogeneity of subjects which can skew the results.
Author Response
The paper is clear, however, methods should be implemented.
Thank you for taking your time to review our paper and for the valuable feedback. We have updated the manuscript in accordance with your comments, please see our response to your questions below.
1) Authors should specify why they selected workers employed in the relocation process.
Response: Thank you for this comment. In the method section we have added information that data collection on self-rated sitting time was performed during to the ongoing measurements in the study evaluating the effects of the relocation process. Page 2, Line 80
“In the original study, the sensor-based measurement of SB and PA were performed twice before relocation (6 and 12 months), and at three timepoints after relocation (6, 11 and 18 months). During the ongoing measurements, we also collected data on self-rated sitting time at the three timepoints after relocation, hereinafter referred to as measurement 1, 2 and 3.”
We also commented on this in the discussion, page 8 Line 276-278.
“The repeated measurements of SB in the study provided a novel possibility to collect parallel data on self-reported SB in the studied group, which is a strength.”
2) In my opinion the age of subjects is very heterogeneous (18-63 years), the population and data analysis should be conducted for age categories (18-35 years; 35-50 years; 50-63 years). The presented data set does not consent to appreciate differences linked to age.
Response: We have added information of the age-span of participants at measurement 1 in the result section. Page 4, Line 173.
Participants were between 28 and 64 years old.
Since our sample size were quite small, it is unfortunately not feasible to conduct age-related analysis. We have however discussed the possible influence of age on self-reports on page 8, Line 273-275.
“Age have previously shown to introduce bias in self-reported SB, but due to the small sample size in our study no age-related analysis was conducted.”
3) what means "working 75% or more"? The total of daily working hours should be specified.
Response: We have clarified the weekly working hours to be included in the study. Page 2, Line 75.
“To be included in the study participants should be 1) 18-63 years of age, 2) working 32 hours a week or more…”
4) What means SF-36? Please clarify.
Response: We have added a sentence describing SF 36. Page 3, Line 100-102.
“Short Form 36 (SF 36) is a questionnaire constructed to survey health status in medical studies.” For assessment of self-rated health, we used one question from SF-36, where participants estimated their health on a five-graded scale from “bad to excellent”.
5) The final population of the study is quite limited, as indicated by the authors in the discussion. This represents the major limitations of the study together with the heterogeneity of subjects which can skew the results.
Response: We have discussed the limitation of the sample size and added a section to discuss the possible effect of age.
Page 8, Line 273.
“Age have previously shown to introduce bias in self-reported SB, but due to the small sample size in our study no age-related analysis was conducted.”
Reviewer 3 Report
This is an elegant study that showed the importance of measuring sedentary time through objective and accurate technologies, rather than simply self-reported questionnaires. The subjectivity of self-administered questionnaires seems to have no more space today with the increasingly accessible technology of accelerometers and inclinometers in monitoring sedentary behavior. It was clear that in a medium-term questionnaire assessment study of negative aspects of life, such as a sedentary lifestyle, people tended to underreport their impressions. The fact that they are small and simple to use, that they are easy to standardize in surveys, and that they drive awareness of the physical activity level outweighs their long-term cost. In addition, the interaction of this data with other devices that measure metabolic rate, heart and respiratory rate, body temperature by infrared technology can make these devices as common and mandatory as the use of smart watches that can alert, guide and identify risk situations for the user look for their healthcare professional.
Author Response
We thank the reviewer for taking the time to read our manuscript and for the valuable comments.
Round 2
Reviewer 2 Report
The paper was improved with some clarifications about the sample and its characteristics. A significant limitation is represented by the limited number of subjects and the absence of stratification for age and sex. However, the methodology adopted is attractive and eligible for publication.